# Textured ferroelectric ceramics with high electromechanical coupling factors over a broad temperature range

Shuai Yang[1], Jinglei Li[1✉], Yao Liu[1], Mingwen Wang[1], Liao Qiao[1], Xiangyu Gao[1], Yunfei Chang [2], Hongliang Du[1], Zhuo Xu[1], Shujun Zhang [3] & Fei Li [1✉]

The figure-of-merits of ferroelectrics for transducer applications are their electromechanical coupling factor and the operable temperature range. Relaxor-$PbTiO_3$ ferroelectric crystals show a much improved electromechanical coupling factor $k_{33}$ (88~93%) compared to their ceramic counterparts (65~78%) by taking advantage of the strong anisotropy of crystals. However, only a few relaxor-$PbTiO_3$ systems, for example $Pb(In_{1/2}Nb_{1/2})O_3$-$Pb(Mg_{1/3}Nb_{2/3})O_3$-$PbTiO_3$, can be grown into single crystals, whose operable temperature range is limited by their rhombohedral-tetragonal phase transition temperatures ($T_{rt}$: 60~120 °C). Here, we develop a templated grain-growth approach to fabricate <001>-textured $Pb(In_{1/2}Nb_{1/2})O_3$-$Pb(Sc_{1/2}Nb_{1/2})O_3$-$PbTiO_3$ (PIN-PSN-PT) ceramics that contain a large amount of the refractory component $Sc_2O_3$, which has the ability to increase the $T_{rt}$ of the system. The high $k_{33}$ of 85~89% and the greatly increased $T_{rt}$ of 160~200 °C are simultaneously achieved in the textured PIN-PSN-PT ceramics. The above merits will make textured PIN-PSN-PT ceramics an alternative to single crystals, benefiting the development of numerous advanced piezo-electric devices.

[1] Electronic Materials Research Laboratory (Key Lab of Education Ministry), State Key Laboratory for Mechanical Behavior of Materials and School of Electronic Science and Engineering, Xi'an Jiaotong University, Xi'an, China. [2] Condensed Matter Science and Technology Institute, School of Instrumentation Science and Engineering, Harbin Institute of Technology, Harbin, China. [3] Institute for Superconducting and Electronic Materials, AIIM, University of Wollongong, Wollongong, NSW, Australia. ✉email: lijinglei@xjtu.edu.cn; ful5@xjtu.edu.cn

Perovskite ferroelectric ceramics, such as lead zirconate titanate (PZT), are the mainstay materials for piezoelectric transducers owing to their high electromechanical properties[1,2]. To optimize the performance of piezoelectric transducers, enlarging the electromechanical coupling factors of ferroelectric materials, which governs the bandwidth, sensitivity, and energy conversion efficiency of the transducers, is of the highest priority[3,4].

Based on previous studies[5,6], the electromechanical coupling factor of perovskite ferroelectrics is inherently associated with the crystallographic orientation. Thus, growing into single crystals is an effective approach to enhance the electromechanical coupling factor of perovskite ferroelectrics. It has been observed that the <001>-oriented rhombohedral $Pb(Mg_{1/3}Nb_{2/3})O_3–PbTiO_3$ (PMN–PT) and $Pb(In_{1/2}Nb_{1/2})O_3–Pb(Mg_{1/3}Nb_{2/3})O_3–PbTiO_3$ (PIN–PMN–PT) (hereafter named relaxor-PT) crystals possess a very high electromechanical coupling factor $k_{33}$ of 88–93%, which is superior to those of state-of-the-art PZT ceramics (65–78%)[7,8]. Due to their enhanced $k_{33}$, PMN–PT crystals have been commercialized in medical imaging transducers, showing greatly broadened bandwidth and enhanced sensitivity when compared to transducers based on PZT ceramics (e.g., one single-crystal transducer was reported to have the ability to cover the frequency range of two PZT ceramic transducers), offering significant advantages in penetration and imaging resolution[9].

However, the main disadvantages of relaxor-PT crystals are the low rhombohedral–tetragonal phase transition temperature $T_{rt}$ (in the range of 60–120 °C) and severe compositional segregation during crystal growth from the melt[10–12]. The low phase transition temperature not only limits the temperature stability of the devices but also affects the fabrication of transducers, since many high-temperature processes, such as packaging and welding, are generally involved in the fabrication process. For example, non-destructive evaluation transducers require temperatures up to 145 °C for operation, while piezoelectric sensors require a temperature of 160 °C, which is much above the $T_{rt}$ of commercially available relaxor-PT crystals[13]. Although various systems with high $T_{rt}$ values, such as $Pb(Mg_{1/3}Nb_{2/3})O_3–PbZrO_3–PbTiO_3$ and $Pb(Sc_{1/2}Nb_{1/2})O_3–PbTiO_3$[14–16], have been designed in recent years, these systems generally contain refractory components, such as $ZrO_2$ and $Sc_2O_3$, leading to difficulties in crystal growth with large size due to their incongruently melting behavior[17,18]. On the other hand, compositional segregation during crystal growth from melts hinders the application of relaxor-PT crystals for large-size piezoelectric transducers, e.g., low-frequency transducers, since it is difficult to obtain large relaxor-PT crystals with acceptable fluctuations in composition and electromechanical properties.

To address the above issues, the fabrication of textured ceramics by the template grain growth (TGG) method was thought to be one of the most promising approaches. TGG involves a solid-state grain growth process, which does not require the melting of material; thus, the negative impacts from refractory oxides on crystal growth can be greatly reduced[19–22]. In recent years, many relaxor-PT solid solutions have been made into textured ceramics[23–27], whose piezoelectric properties are close to those of relaxor-PT crystals grown from melts. However, the $T_{rt}$ of those textured ceramics does not show obvious advantages over state-of-the-art relaxor-PT crystals since the presence of $BaTiO_3$ (BT) or $SrTiO_3$ (ST) templates have been found to greatly lower the $T_{rt}$. For example, the $T_{rt}$ of 0.4PMN–0.25PZ–0.35PT decreases from 160 to 75 °C by adding 5 vol.% BT templates[27], while the $T_{rt}$ of $0.16Pb(Yb_{1/2}Nb_{1/2})O_3–0.52Pb(Mg_{1/3}Nb_{2/3})O_3–0.32PbTiO_3$ was found to decrease from 120 °C to approximately room temperature by adding 3 vol.% BT templates[24]. One may argue that the $T_{rt}$ of PMN–PZ–PT solid solution can be further increased by

increasing the PZ content, however the high level of PZ will cause difficulties in template-induced grain growth. Currently, the highest PZ content for the PMN–PZ–PT that can be textured was found to be around 25%[27].

## Results

In this work, we used the TGG method to fabricate <001>-textured $0.19Pb(In_{1/2}Nb_{1/2})O_3–xPb(Sc_{1/2}Nb_{1/2})O_3–(0.81−x)PbTiO_3$ (PIN–PSN–PT) ceramics ($x = 0.44–0.49$), since our previous research found that the $T_{rt}$ of this solid solution was in the range of 210–240 °C[28,29], offering more freedom for $T_{rt}$ tailoring by the templates. The <001>-oriented BT microplates were synthesized, and the length and thickness were approximately 7 and 0.6 μm, respectively (Supplementary Fig. 1), with an aspect ratio on the order of ten, which is appropriate for use as templates in TGG. The BT templates were aligned in the PIN–PSN–PT matrix by using the tape casting technique. Due to the high melting temperature of PIN–PSN–PT (>1400 °C), template-induced PIN–PSN–PT grain growth was minimal even at a high temperature of 1250 °C (Fig. 1a). Further increasing the temperature, however, led to the significant volatilization of PbO, thus greatly deteriorating the densification and properties of the final samples. To resolve this issue, various sintering aids, including PbO, $Li_2CO_3$, $B_2O_3$, and CuO, were applied to assist template-induced grain growth. The results showed that a small amount of CuO and $B_2O_3$ (0.2–0.5 wt%) could effectively facilitate the growth of PIN–PSN–PT single crystals on BT templates, as shown in Fig. 1b. Centimeter-scale textured samples are shown in Fig. 1c. Thus, in the following text, all textured PIN–PSN–PT ceramics were sintered with the addition of CuO and $B_2O_3$.

Figure 1d shows XRD patterns of 0.19PIN–0.445PSN–0.365PT ceramics textured with 3, 5, and 7 vol.% BT templates and compared to their nontextured counterparts. All samples have a pure perovskite structure, while the textured ceramics exhibit greatly enhanced intensities of the (001) peaks, in which the Lotgering factor $F_{001}$ is found to be ~99% for all textured ceramics. Electron backscatter diffraction (EBSD) experiments further revealed that most grains are well oriented along the <001> direction for the textured PIN–PSN–PT ceramics, as shown in Fig. 1e, f. In respect to the microstructure, the largest difference among the textured ceramics with different volume fractions of templates is the grain size. As shown in Supplementary Fig. 2, it is observed that the grain size decreases from 22 to 14 μm by increasing the concentration of BT templates from 3 to 7 vol.%, which is consistent with the relationship between the spacing and the number frequency of the templates, i.e., $x_T = \left(\frac{6}{\pi f_T}\right)^{1/3}$, where $x_T$ is the spacing of the templates and $f_T$ is the number frequency of the templates[20]. It should be noted here that the grain size of the textured ceramic is the same as $x_T$ if the ceramic is completely textured.

To characterize the phase transition temperature $T_{rt}$ of the textured 0.19PIN–0.445PSN–0.365PT ceramic, the X-ray diffraction (XRD) pattern and dielectric constant were measured as a function of temperature (Fig. 2). In XRD experiments, $T_{rt}$ can be determined from the splitting of the (002) diffraction peak. For nontextured counterparts, however, the splitting of the (002) diffraction peak is not obvious from room temperature to the Curie temperature (Fig. 2a), which is attributed to the fact that $T_{rt}$ is very close to the Curie temperature. In this case, the lattice parameter of the high-temperature tetragonal phase is very close to that of the cubic phase; therefore, the resolution of the present XRD experiment is not sufficient to detect the splitting of the (002) peak. According to the temperature-dependent dielectric constant, the $T_{rt}$ of nontextured ceramics is found to be ~210 °C, as shown in Fig. 2e. This result is also confirmed by the

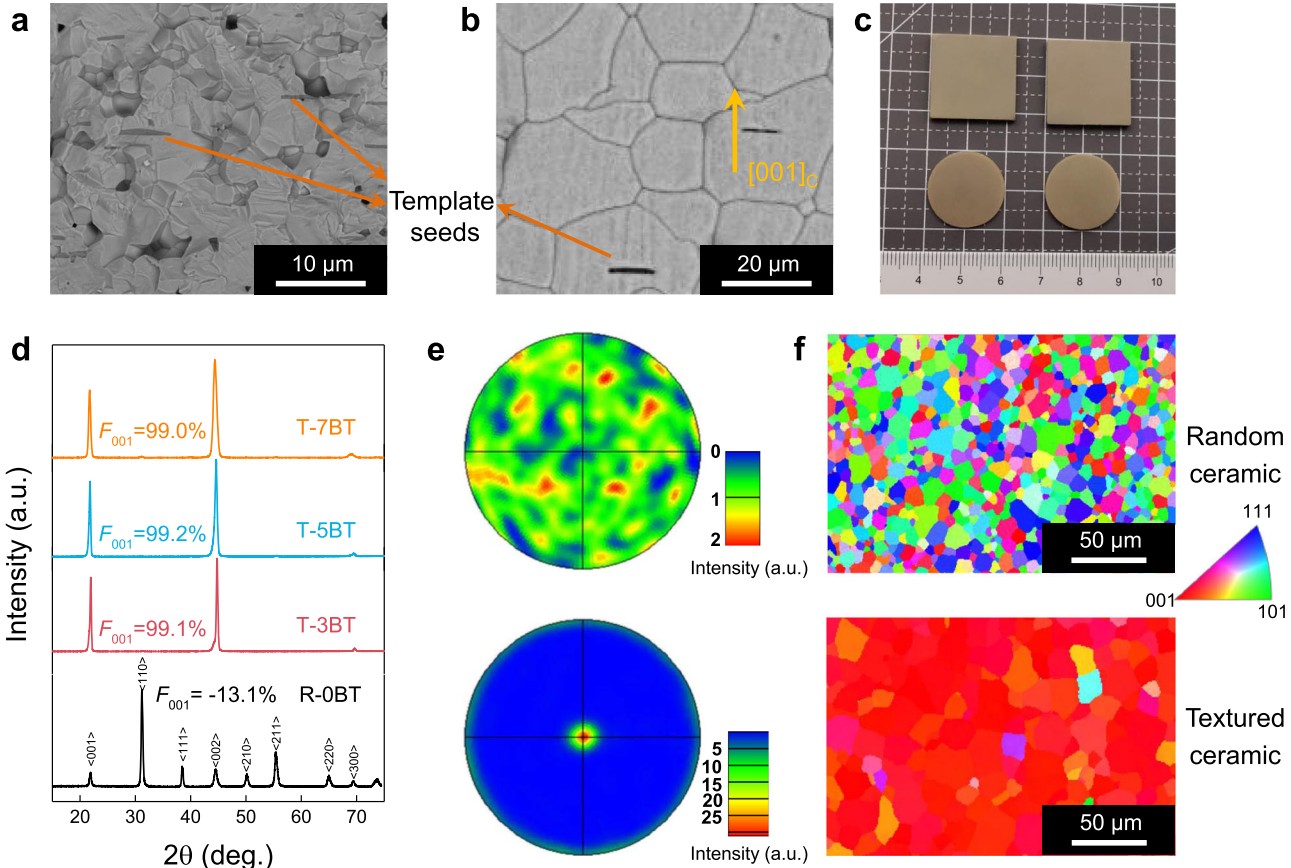

**Fig. 1 Microstructure of BT templates and 0.19PIN–0.445PSN–0.365PT textured ceramics. a** SEM image of textured PIN–PSN–PT ceramic without sintering aids (fracture surface). **b** SEM image of textured PIN–PSN–PT ceramic with CuO and $B_2O_3$ as sintering aids (fracture surface). **c** Picture of textured PIN–PSN–PT ceramic with CuO and $B_2O_3$ as sintering aids. **d** XRD patterns of the random ceramic and the textured ceramics with 3, 5, and 7 vol.% BT templates, the $F_{001}$ of the random ceramic and textured ceramics were calculated based on the XRD pattern of the PIN–PSN–PT powders. **e** The <001> pole figures of the nontextured ceramic (top) and the 3 vol.% BT-textured ceramic (bottom), where the color bar indicates the intensity in arbitrary units. **f** Grain orientation images (measured by the SEM-EBSD technique on sample surface) of the random ceramic and the 3 vol.% BT-textured ceramic, where the colors indicate the orientation of the grains.

temperature dependence of the (222) diffraction pattern (Supplementary Fig. 3), where the splitting of the (222) diffraction peak disappears in the temperature range of 200–220 °C for the nontextured sample. For the 3, 5, and 7 vol.% BT-textured 0.19PIN–0.445PSN–0.365PT ceramics, the $T_{rt}$ values were found to be approximately 170, 110, and 50 °C, respectively, according to the (002) diffractions (Fig. 2b–d). It should also be noted here that the 7 vol.% BT-textured sample exhibits an obviously asymmetric (002) peak even at room temperature, revealing a mixed phase state at room temperature. As shown in Fig. 2f–h, the $T_{rt}$ values observed in the temperature-dependent dielectric constant measurements are basically the same as those observed in the XRD experiments for the 3, 5, and 7 vol.% BT-textured PIN–PSN–PT ceramics. In contrast to $T_{rt}$, the Curie temperature $T_c$ of the 0.19PIN–0.445PSN–0.365PT ceramics is not sensitive to the BT templates and slightly decreases from 260 to 240 °C by adding BT templates up to 7–vol.%, as shown in Fig. 2e–h. The decrease in $T_{rt}$ with increasing BT templates can be attributed to the following two factors. First, the presence of BT templates in the PIN–PSN–PT matrix favors the tetragonal phase since BT is in the tetragonal phase from room temperature to 120 °C. Second, the diffusion of BT into the PIN–PSN–PT matrix may also reduce the $T_{rt}$ of the textured ceramics, as observed in Ba-doped relaxor-PTs (Supplementary Fig. 4). Based on phase-field simulations

(Fig. 3), $T_{rt}$ is found to decrease by only 10 °C by adding 5vol.% BT into the PMN–30PT matrix (without the diffusion of BT in the matrix). Thus, it is thought that the relatively large decrease in $T_{rt}$ in the textured PIN–PSN–PT ceramics is mainly attributed to the diffusion of BT in the PIN–PSN–PT matrix, which was observed by Scanning Electron Microscope-Energy Dispersive Spectrometer (SEM-EDS) experiments, as shown in Supplementary Fig. 5.

The electromechanical-related properties of the textured 0.19PIN–0.445PSN–0.365PT ceramics are shown in Fig. 4 and Table 1. It is worth noting the electromechanical coupling factors $k_{33}$s of textured ceramics were measured by three different methods, as described in the "Methods" section. Figure 4b gives the measurement data of $k_{33}$ from a longitudinal bar of the 3 vol.% BT-textured PIN–PSN–PT ceramic. The electromechanical coupling factors are similar for the textured ceramics with different volume fractions of BT templates since the texture quality is almost the same for these ceramics (Fig. 1d). The coupling factors $k_{33}$, $k_{31}$, and $k_p$ of the textured 0.19PIN–0.445PSN–0.365PT ceramics are found to be approximately 87–90%, 55–58%, and 80–83%, respectively, which are much larger than those of conventional PZT ceramics (Fig. 4a and Table 1) and comparable to those of relaxor-PT crystals. It should be noted that the relaxor-PT crystal does not possess the

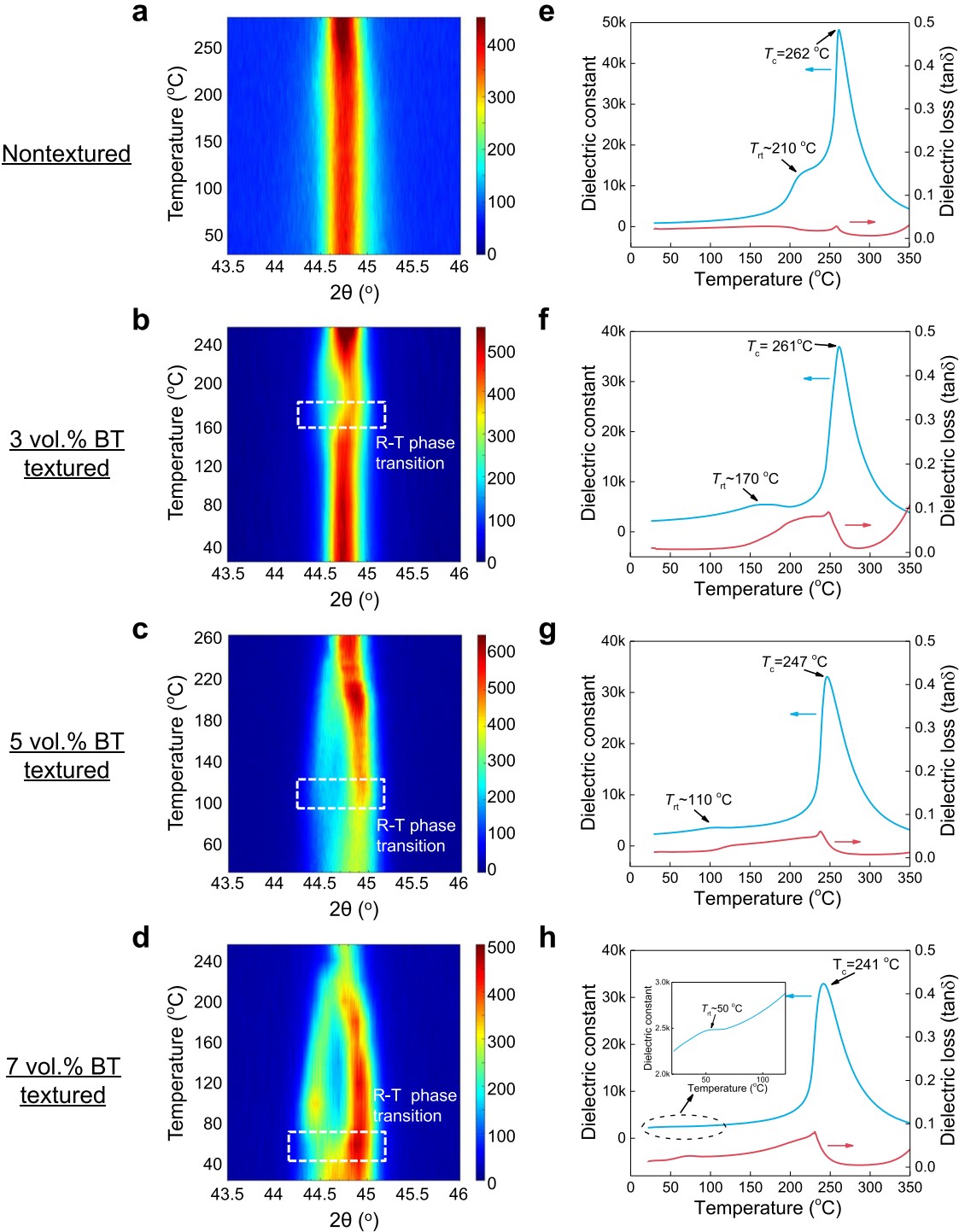

**Fig. 2 Determination of rhombohedral-tetragonal phase transition temperature ($T_{rt}$) of nontextured and textured 0.19PIN–0.445PSN–0.365PT ceramics. a–d** The <002> X-ray diffractions as a function of temperature for nontextured and textured PIN–PSN–PT ceramics. The colors indicate the intensity of X-ray diffractions. **e–h** The temperature dependence of dielectric behavior at 1 kHz for PIN–PSN–PT ceramics.

planar electromechanical coupling factor $k_p$ since its macroscopic symmetry is 4 mm; thus, the crystal plate exhibits in-plane elastic anisotropy. The macroscopic symmetry of a textured ceramic, however, is the same as that of conventional ceramics, i.e., $\infty m$; thus, the radial vibration mode is still available for textured ceramics. In contrast to coupling factors, the piezoelectric coefficient $d_{33}$ is more sensitive to the volume fraction of BT templates. The presence of BT templates leads to two opposite effects on the piezoelectric coefficient. First, the piezoelectricity of the

<001>-oriented BT template is smaller than that of the <001>-textured PIN–PSN–PT matrix, leading to a decrease in $d_{33}$. Second, BT templates decrease the $T_{rt}$ (as observed in Fig. 2) of textured PIN–PSN–PT, which may result in an increased room-temperature dielectric constant and piezoelectric coefficient. In this work, the highest $d_{33}$ is observed with the 5 vol.% BT-templated 0.19PIN–0.445PSN–0.365PT ceramic, which is ~1100 pC N$^{-1}$; this value is double the value of soft PZT ceramics with a similar Curie temperature (Table 1).

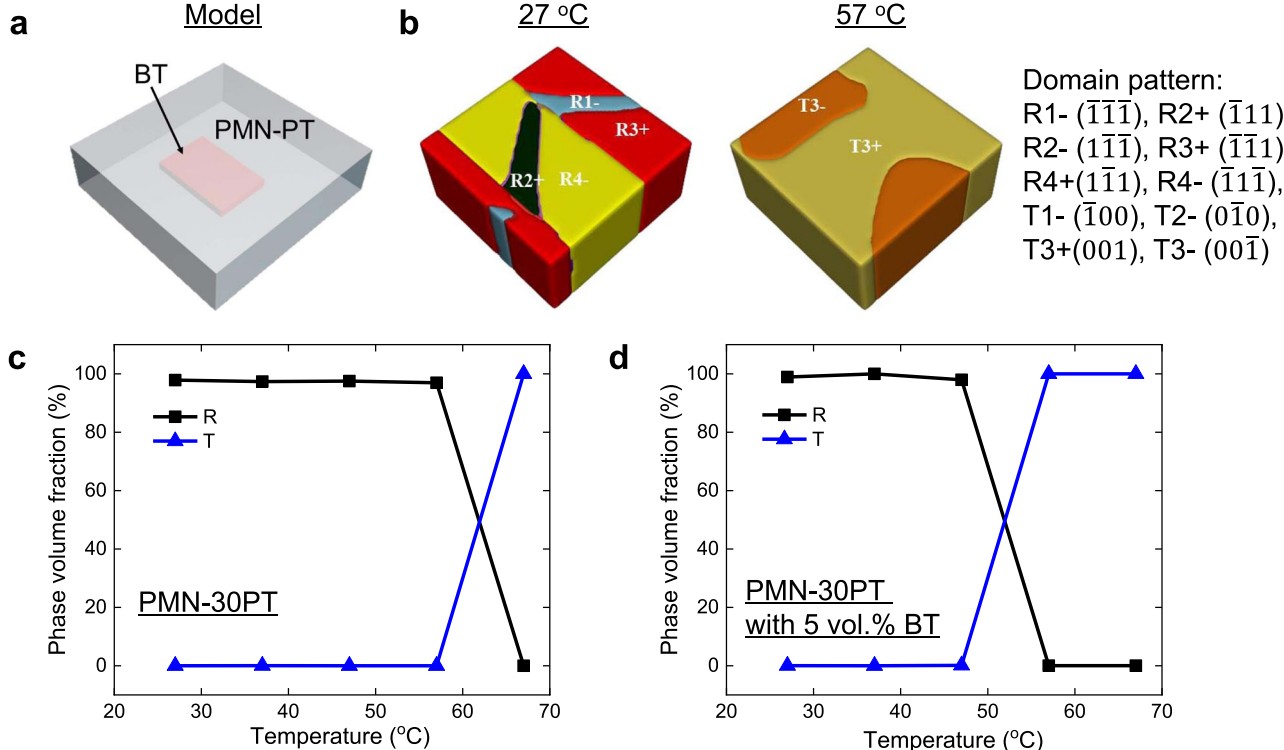

**Fig. 3 Phase-field simulation of domain structure for PMN–30PT and PMN–30PT with a BT template. a** Three-dimensional schematic illustration of PMN–30PT with a BT plate in the simulation. **b** Simulated domain structures of PMN–30PT with a BT plate (5 vol.%) at 27 °C and 57 °C, where the colors indicate different ferroelectric domains (the corresponding polar directions are given in the right of the figure). **c** Simulated phase volume fraction for PMN–30PT. **d** Simulated phase volume fraction for PMN–30PT with 5 vol.% BT template.

The electric field-induced strains for the textured 0.19PIN–0.445PSN–0.365PT ceramics are shown in Fig. 4c. The highest electric field-induced strain is observed in the textured ceramic with 3 vol.% BT templates, being 0.38% at an electric field of 3 kV mm$^{-1}$ (the general working electric field for piezoelectric actuators), which is comparable to that of a commercial PMN–PT crystal (see Fig. 4d) but three times larger than that of its non-textured counterpart and double the value of state-of-the-art PZT ceramics (~0.2%)[30]. The 7 vol.% BT-textured sample exhibits an inferior strain level when compared to its counterparts with lower template volume fractions because the strain in the <001>-tex-tured rhombohedral PIN–PSN–PT ceramics is mainly from the electric field-induced rhombohedral–tetragonal phase transition[7,10], while the presence of BT templates favors the tet-ragonal phase and decreases the strain contribution of the rhombohedral-to-tetragonal phase transition. It should be noted here that a low strain hysteresis is expected for an ideally textured rhombohedral perovskite ceramic, as observed in the PMN–28PT crystal, since only four degenerated ferroelectric domains along the [111], [1$\bar{1}$1], [$\bar{1}$11], and [$\bar{1}$1$\bar{1}$] directions exist in [001]-poled rhombohedral grains, which are energetically stable under the application of a [001] electric field. In regard to actual textured ceramics, however, the grains are not exactly along the <001> direction and are responsible for the large strain hysteresis (~10%), as observed in Fig. 4d. This phenomenon is evidenced in the SEM image (Supplementary Fig. 6, the large surfaces of the BT templates are not exactly parallel to the sample surface and each other) and the XRD rocking curves of the textured ceramics (Supplementary Fig. 7), where the full-width-half-maximum (FWHM) is 7°; this FWHM value is much larger than that of a single crystal (~1°). It is expected that the strain hysteresis can be

decreased by improving the texturing quality to reduce the orientation fluctuation of the templates in the textured ceramics.

Figure 4e shows the polarization-electric field (PE) loops for the textured 0.19PIN–0.445PSN–0.365PT ceramics. The coercive fields are almost the same for the textured and nontextured ceramics, with a value of 7.5 kV cm$^{-1}$; this value is much larger than those of <001>-oriented rhombohedral PMN–PT (2.5 kV cm$^{-1}$) and PIN–PMN–PT crystals (4 kV cm$^{-1}$). As the coercive field is proportional to the Curie temperature[10], the higher coercive field of PIN–PSN–PT is thought to be associated with its higher Curie temperature when compared to the PMN–PT and PIN–PMN–PT systems. In regard to high-power transducer applications, a large coercive field is highly desired, which guarantees that ferroelectric materials are not depolarized under a high driving electric field[3,31]. In addition, it can be seen from Fig. 4e that the remnant polarization of textured ceramics (0.31 C m$^{-2}$) is clearly lower than that of its nontextured coun-terpart (0.40 C m$^{-2}$), again demonstrating the high Lotgering factor of these <001>-texture ceramics. For perovskite rhombo-hedral ferroelectric crystals, the lowest remnant polarization is along the <001>-direction because the spontaneous polarizations are along the eight <111> crystallographic directions[3,10]. Thus, the decrease in remnant polarization indicates an increased alignment of the <001>-oriented grains.

From an application viewpoint, the piezoelectric and electro-mechanical properties of the textured ceramics were studied as a function of temperature, as shown in Fig. 5. As expected, the textured ceramics do not show property degradation at the temperature below their respective $T_{rt}$s (Fig. 5a, b), leading to a monotonous increase in $d_{33}$ from room temperature to $T_{rt}$. It is worth noting here that the electric-field-induced strain also

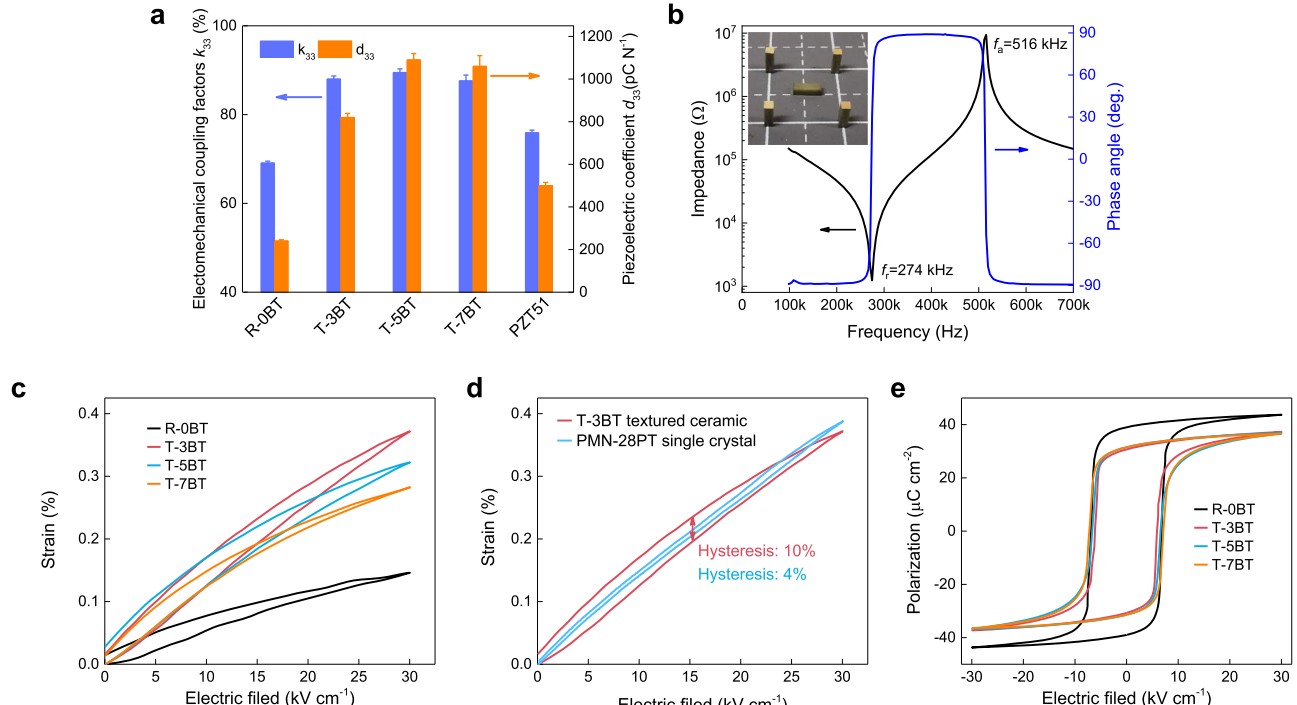

**Fig. 4 The electromechanical properties of textured 0.19PIN–0.445PSN–0.365PT ceramics. a** The $k_{33}$ and $d_{33}$ values for nontextured (R-0BT), 3 vol.% BT-textured (T-3BT), 5 vol.% BT-textured (T-5BT), and 7 vol.% BT-textured (T-7BT) 0.19PIN–0.445PSN–0.365PT ceramics, compared with commercially available "soft" PZT ceramic, which has the similar Curie temperature (PZT51, Yu Hai Electronic Ceramics Co., Ltd., China). Five samples for each component are used for the characterization of $d_{33}$ and $k_{33}$. The error bars present the standard deviation of the corresponding data. **b** Impedance/phase spectra for the longitudinal mode of a T-3BT 33-bar. The inset shows the picture of T-3BT 33-bars ($4 \times 1 \times 1\,mm^3$). **c** The electric-field-induced strains for R-0BT, T-3BT, T-5BT, and T-7BT ceramics. **d** A comparison of the electric-field-induced strains between the T-3BT ceramic and <001>-poled PMN–28PT single crystal. **e** The polarization-electric field (PE) loops of R-0BT, T-3BT, T-5BT, and T-7BT ceramics. The strain hysteresis ($H_s$) is defined as $\Delta S_{E_{max}/2}/S_{max} \times 100\%$, where $\Delta S_{E_{max}/2}$ and $S_{max}$ are the strain difference with rising and falling fields at half maximum electric field ($E_{max}/2$) and the strain at maximum electric field ($E_{max}$), respectively.

**Table 1 Electromechanical properties of nontextured and textured 0.19PIN–0.445PSN–0.365PT ceramics.**

| Samples | $d_{33}$ (pC N$^{-1}$) | $\varepsilon_{33}^r$ | tan$\delta$ (%) | $T_{rt}$ (°C) | $T_c$ (°C) | $k_{33}$ (%) | $k_{31}$ (%) | $k_p$ (%) | $k_t$ (%) |
|---|---|---|---|---|---|---|---|---|---|
| R-0BT | 240 ± 10 | 860 ± 20 | 2.3 | 210 | 262 | 69 ± 0.4 | 27.3 ± 0.2 | 51.5 ± 0.3 | 52.3 ± 0.3 |
| T-3BT | 850 ± 30 | 2120 ± 60 | 0.8 | 170 | 261 | 88.0 ± 0.5 | 54.5 ± 0.4 | 82.2 ± 0.7 | 59.9 ± 0.5 |
| T-5BT | 1090 ± 50 | 2310 ± 90 | 1.2 | 110 | 247 | 89.5 ± 0.7 | 57.7 ± 0.5 | 83.4 ± 0.7 | 61.6 ± 0.6 |
| T-7BT | 1060 ± 50 | 2250 ± 110 | 1.3 | 50 | 241 | 86.9 ± 0.4 | 58.1 ± 0.5 | 80.5 ± 0.5 | 55.9 ± 0.6 |
| PZT51 | 500 | 2250 | 2.0 | / | 259 | 75 | 38 | 64 | 52 |
| PMN–PT crystal | 1500 | 5000 | 0.5 | 92 | 130 | 91 | 58 | / | 62 |

The R-0BT, T-3BT, T-5BT, and T-7BT indicate random, 3 vol.% BT-textured, 5 vol.% BT-textured, and 7 vol.% BT-textured 0.19PIN–0.445PSN–0.365PT ceramics. The data of a commercially available "soft" PZT (PZT51, Yu Hai Electronic Ceramics Co., Ltd., China) ceramic and a PMN-0.28PT crystal was also given for comparison. $k_{33}$: longitudinal-mode coupling factor; $k_{31}$: lateral-mode coupling factor; $k_p$: planar-mode coupling factor; $k_t$: thickness-mode coupling factor. For each composition, five samples were used for the measurements of electromechanical properties and the standard errors were given in the table.

increases monotonously from room temperature to $T_{rt}$, which is similar to the change of $d_{33}$, as shown in Supplementary Fig. 8. Since the piezoelectric coefficient, dielectric permittivity and elastic compliance all increase from room temperature to $T_{rt}$, the electromechanical coupling factor $k_{33}$ is expected to remain stable over this temperature range according to the following equation, $k_{33} = d_{33}/\sqrt{\varepsilon_{33}^T s_{33}^E}$ (where $\varepsilon_{33}^T$ is the free dielectric permittivity and $s_{33}^E$ is the elastic compliance coefficient under short circuit). Of particular interest is that the 3 vol.% BT-textured PIN–PSN–PT sample possesses a very high coupling factor of 88% up to 160 °C. This operable temperature range is much larger than state-of-the-art relaxor-PT crystals, as shown in Fig. 5c, d. It is worth noting here that the operable temperature range ($T_{rt}$) of

textured PIN–PSN–PT ceramics can be further enlarged by tuning the composition, i.e., reducing the PT content or the volume fraction of BT templates. We fabricated a series of <001>-textured 0.19PIN–xPSN–(0.81−x)PT ceramics with x in the range of 0.44–0.49, whose electromechanical coupling factor $k_{33}$ and $T_{rt}$ are shown in Fig. 5d (see Table 2 for detailed properties). It can be seen that $T_{rt}$ increases up to 200 °C with a $k_{33}$ of 85% by modifying the composition. The slight decrease in $k_{33}$ with increasing $T_{rt}$ is due to the following two factors. First, the decrease in BT content may affect the texturing degree of the ceramics, i.e., some grains are not textured along the <001> direction, as shown in Supplementary Fig. 9. Second, the decrease in PT content moves the composition away from the

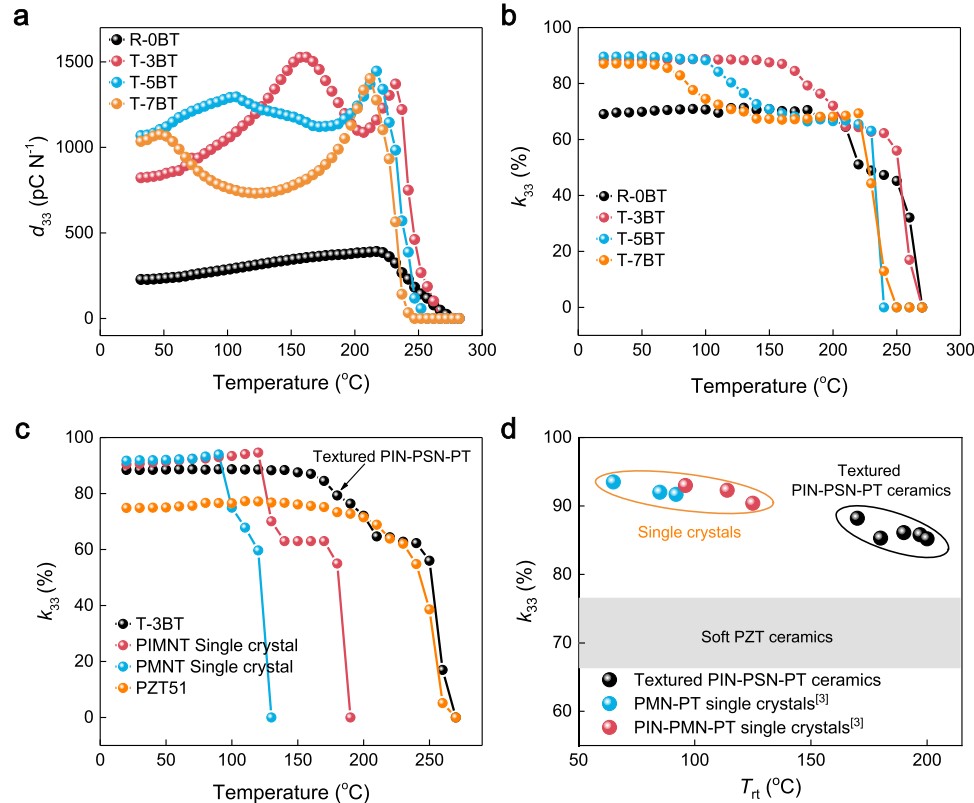

**Fig. 5 The temperature dependence of electromechanical properties of textured PIN–PSN–PT ceramics. a** The temperature dependence of piezoelectric coefficient $d_{33}$ for nontextured (R-0BT), 3 vol.% BT-textured (T-3BT), 5 vol.% BT-textured (T-5BT), 7 vol.% BT-textured (T-7BT) 0.19PIN–0.445PSN–0.365PT ceramics. **b** The temperature dependence of electromechanical coupling factor $k_{33}$ for R-0BT, T-3BT, T-5BT, and T-7BT ceramics. **c** A comparison of the temperature dependence of $k_{33}$ among T-3BT-textured ceramic, "soft" PZT ceramic (PZT51) and <001>-oriented relaxor-PT single crystals (data of relaxor-PT crystals from ref. [10]). **d** The $k_{33}$ as a function of $T_{rt}$ for a series textured PIN–PSN–PT ceramics and relaxor-PT single crystals, where the $k_{33}$ values of state-of-the-art "soft" PZT ceramics are marked in the figure by the gray color.

**Table 2 Electromechanical properties of textured PIN–PSN–PT ceramics.**

| PIN–PSN–PT composition | | | BT (vol.%) | $d_{33}$ (pC N$^{-1}$) | $\varepsilon^r_{33}$ | $T_{rt}$ (°C) | $T_c$ (°C) | $k_{33}$ (%) |
|---|---|---|---|---|---|---|---|---|
| **PIN** | **PSN** | **PT** | | | | | | |
| 0.19 | 0.445 | 0.365 | 1 | 620 ± 20 | 1530 ± 50 | 197 | 262 | 85.8 ± 0.5 |
| 0.19 | 0.445 | 0.365 | 2 | 700 ± 30 | 1650 ± 70 | 190 | 261 | 86.1 ± 0.3 |
| 0.19 | 0.445 | 0.365 | 3 | 850 ± 30 | 2120 ± 60 | 170 | 261 | 88.0 ± 0.5 |
| 0.19 | 0.445 | 0.365 | 4 | 930 ± 60 | 2160 ± 100 | 140 | 258 | 88.0 ± 0.6 |
| 0.19 | 0.445 | 0.365 | 5 | 1090 ± 50 | 2310 ± 90 | 110 | 248 | 89.5 ± 0.7 |
| 0.19 | 0.465 | 0.345 | 3 | 670 ± 30 | 1600 ± 70 | 180 | 243 | 85.3 ± 0.2 |
| 0.19 | 0.485 | 0.325 | 3 | 600 ± 20 | 1390 ± 40 | 200 | 231 | 85.2 ± 0.3 |

For each composition, five samples were used for the measurements of electromechanical properties and the standard errors were given in the table.

morphotropic phase boundary, thus leading to a decrease in piezoelectric and electromechanical properties.

## Discussion

In summary, we addressed a challenge that has been present in relaxor-PT ferroelectrics, i.e., the large impact of the low ferroelectric phase transition temperature of relaxor-PT crystals on the design freedom of high-performance piezoelectric devices. Compared to state-of-the-art relaxor-PT crystals, the <001>-textured PIN–PSN–PT ceramics exhibit not only comparable electromechanical coupling factors ($k_{33}$) of 85–88% and electric field-induced strains of 0.38% (at 3 kV mm$^{-1}$) but also enhanced $T_{rt}$ values of 160–200 °C and coercive field of 7.5 kV cm$^{-1}$, greatly broadening their operable temperature range and drive field stability. In addition to the merits above, the high composition uniformity thus high property homogeneity, and low manufacturing cost compared to single crystals grown from melt, together with the availability of the radial vibration mode, make the <001>-textured PIN–PSN–PT ceramics beneficial in numerous advanced piezoelectric transducers and sensors. Finally, it is worth noting that there is still space to further enhance the electromechanical properties of the textured PIN–PSN–PT ceramics by improving the fabrication process. For instance, the density of the studied textured ceramics was only 95% (Supplementary Table 1). This value could be improved by hot isotropic press sintering, thus leading to higher electromechanical properties in the resulting textured ceramics.

## Methods

**Synthesis of <001>-oriented BT template.** The BT templates were synthesized by two-step topological chemical reaction according to the following formulas:

$$2Bi_2O_3 + 3TiO_2 = Bi_4Ti_3O_{12} \tag{1}$$

$$Bi_4Ti_3O_{12} + 3BaCO_3 = 3BaTiO_3 + 2Bi_2O_3 + 3CO_2 \tag{2}$$

First, $Bi_2O_3$ and $TiO_2$ powders were selected as raw materials, NaCl and KCl were used as molten salt to prepare the bismuth based lamellar structure, $Bi_4Ti_3O_{12}$ platelets. Second, the precursor $Bi_4Ti_3O_{12}$/$BaCO_3$ with a molar ratio of 1:10 were mixed with the salt (NaCl and KCl with a molar ratio of 1:1) and sintered at 1020 °C for 2 h, and then washed with hot deionized water and dilute acid to obtain the <001> BT templates.

**Synthesis of PIN–PSN–PT powder.** The PIN–PSN–PT powder was prepared by the conventional high-temperature solid-state reaction method. The $Pb_3O_4$, $InNbO_4$, $ScNbO_4$, and $TiO_2$ powders were wet mixed by ball milling for 6 h and then calcined at 850 °C for 2 h to achieve the pure perovskite phase with nominal compositions. In order to obtain a large size difference between the template and matrix particles for assisting TGG[20,32], the powder was ball milled for 72 h before tape casting and the average powder size was 340 nm, as shown in Supplementary Fig. 10.

**Fabrication of textured ceramics.** For tape casting, the slurry was prepared by mixing the PIN–PSN–PT powders and BT templates with ethanol/xylene co-solvents, KD-1 dispersant, polyvinyl butyral (PVB) binder, butyl benzyl phthalate (BBP)/polyalkylene glycol (PAG) plasticizer, and CuO and $B_2O_3$ (0.2–0.5 wt%) liquid flux. The slurry was tape casted at a rate of 0.5 cm s$^{-1}$ using a stainless-steel blade with the thickness of 200 μm. After drying, green tapes were cut, stacked, and uniaxial pressed under 20 MPa pressure at 70 °C for 10 min. The binder was burnt out at 600 °C for 2 h with a heating rate of 0.5 °C min$^{-1}$. Finally, the textured samples were sintered at 1200–1250 °C for 10 h to ensure the sufficient grain growth, while samples without BT platelets were sintered at 1250 °C for 2 h. Density was measured by Archimedes method, as listed in Supplementary Table 1.

**Characterization of structure, morphology, and textured degree.** The phase structure was measured by XRD (SmartLab, Japan), while the textured degree was determined by the Lotgering method and rocking curve. For Lotgering method, the texturing degree $F_{001}$ was estimated by the following formula:

$$F_{00l} = \frac{P - P_0}{1 - P_0}, P = \frac{\sum I_{(00l)}}{\sum I_{(hkl)}}, P_0 = \frac{\sum I_{0(00l)}}{\sum I_{0(hkl)}} \tag{3}$$

where $I$ and $I_o$ are the X-ray intensities of textured and random samples, respectively.

To determine the phase transition temperature, the ceramic samples were ground into powder and the XRD patterns were measured every 10 °C from room temperature to 280 °C.

Field-emission scanning electron microscope (FE-SEM, Quanta F250, FEI, USA) was used to characterize the microstructure of the ceramics. The EBSD technique was employed to measure the distribution of grain orientation of the samples.

**Electromechanical properties measurements.** Before electromechanical measurements, all the samples were poled by an electric field of 3 kV mm$^{-1}$ in silicone oil for 10 min at room temperature. The $d_{33}$ values were determined by a quasi-static $d_{33}$-meter (ZJ-4A, Institute of Acoustics, China). The temperature dependence of $d_{33}$ value was measured by an in situ measuring instrument (TZFD-600, Harbin Julang Technology Co.Ltd, China) based on quasi-static method. The dielectric constant ($\varepsilon_{33}^T$) and dielectric loss (tanδ) were measured as a function of temperature using a LCR meter (E4980A, Agilent, Palo Alto, USA) being connected to a computer-controlled furnace. The electromechanical coupling factors were characterized by resonance–antiresonance method using an impedance analyzer (HP4294A, Agilent, USA). The sizes of the samples are $\Phi10 \times 1 \ mm^3$ and $10 \times 3 \times 1 \ mm^3$ for planar $k_p$, thickness $k_t$, and lateral $k_{31}$ measurements, respectively. To measure the coupling factor $k_p$, the Poisson's ratio of the textured ceramics was first determined by the ratio of first overtone to fundamental resonant frequencies of radial vibration mode[33]. For textured PIN–PSN–PT ceramics, the Poisson's ratio ($-s_{12}^E/s_{11}^E$) is in the range of $-0.025$–$0.2$ (depends on the composition), much smaller than that of nontextured counterpart (0.37), being consistent with previous observation[25]. Then, the $k^p$ (planar radial piezoelectric coupling coefficient) was determined by σ and $(f_a - f_r)/f_r$ based on the ref. [33] Finally, the coupling factor $k_p$ was calculated by Eq. (4). All the parameters of textured 0.19PIN–0.445PSN–0.365PT ceramics in the process of calculating $k_p$ are listed in

Supplementary Table 2.

$$k_p^2 = \frac{(k^p)^2}{[((1+\sigma)/2) + (k^p)^2]} \tag{4}$$

The coupling factors $k_t$ and $k_{31}$ were calculated by the following formulas following IEEE Standard on Piezoelectricity:

$$\frac{k_{31}^2}{1 - k_{31}^2} = \frac{\pi f_a}{2 f_r} \tan\left(\frac{\pi}{2} \frac{f_a - f_r}{f_r}\right) \tag{5}$$

$$k_t^2 = \frac{\pi f_r}{2 f_a} \tan\left(\frac{\pi}{2} \frac{f_a - f_t}{f_a}\right) \tag{6}$$

where $f_r$ and $f_a$ are the resonant frequency and anti-resonant frequency, respectively.

Three methods were used to determine the longitudinal coupling $k_{33}$: (1) based on the factors $k_p$ and $k_t$, as shown in the empirical Eq. (7); (2) based on the free and clamped dielectric permittivity, as shown in Eq. (8); (3) measured from the 33-bars with size of $4 \times 1 \times 1 \ mm^3$, by using Eq. (9) according to IEEE Standard on Piezoelectricity.

$$k_{33}^2 \approx k_p^2 + k_t^2 - k_p^2 \times k_t^2 \tag{7}$$

$$k_{33} = \sqrt{1 - \frac{\varepsilon_{33}^S}{\varepsilon_{33}^T}} \tag{8}$$

$$k_{33}^2 = \frac{\pi f_r}{2 f_a} \tan\left(\frac{\pi}{2} \frac{f_a - f_r}{f_a}\right) \tag{9}$$

where $\varepsilon_{33}^S$ is the clamped dielectric permittivity, $\varepsilon_{33}^T$ the free dielectric permittivity, $f_r$ the resonant frequency, and $f_a$ the anti-resonant frequency. The $k_{33}$ calculated by the three methods are very similar with a variation below 2%. The temperature dependence of these coupling factors was measured by connecting the impedance analyzer (HP4294A) to a computer-controlled furnace.

PE hysteresis loops and strain-electric field (SE) curves were measured using a ferroelectric testing system (TF Analyzer 2000, aix-ACCT, Aachen, Germany), being connected with a laser interferometer vibrometer (SPeS 120, SIOS Mebtechnik GmbH, Germany).

**Phase-field simulations.** To describe the impact of BT templates on the phase transition temperature of rhombohedral PMN–PT, a model with a BT microplate surrounded by PMN–30PT matrix was employed, as shown in Fig. 3a. The volume fraction of BT template was 5 vol.% in the system. In phase-field simulation, the time-dependent Ginzburg-Landau (TDGL) equation was used to exhibit the temporal evolution of the polarization for the system[34],

$$\frac{\partial P_i(\mathbf{r}, t)}{\partial t} = -L \frac{\delta F}{\delta P_i(\mathbf{r}, t)} (i = 1, 2, 3) \tag{10}$$

where $L$ is the kinetic coefficient, $F$ the total free energy of the system, $r$ the space position, and $P_i(\mathbf{r}, t)$ is the polarization. The total free energy of the system can be expressed as:

$$F = \int_V [f_{bulk} + f_{elas} + f_{elec} + f_{grad}] dV \tag{11}$$

where $f_{bulk}$, $f_{elas}$, $f_{elec}$, and $f_{grad}$ represent the Landau bulk free energy density, the elastic energy density, the electrostatic energy density, and the gradient energy density, respectively, $V$ is the system volume. The sixth-order bulk free energy can be expressed as:

$$
\begin{aligned}
f_{bulk} =\ & \alpha_1 \left(P_1^2 + P_2^2 + P_3^2\right) + \alpha_{11}\left(P_1^4 + P_2^4 + P_3^4\right) + \alpha_{12}\left(P_1^2 P_2^2 + P_2^2 P_3^2 + P_3^2 P_1^2\right) \\
& + \alpha_{111}\left(P_1^6 + P_2^6 + P_3^6\right) + \alpha_{112}\left[P_1^4\left(P_2^2 + P_3^2\right) + P_2^4\left(P_1^2 + P_3^2\right) + P_3^4\left(P_1^2 + P_2^2\right)\right] \\
& + \alpha_{123} P_1^2 P_2^2 P_3^2
\end{aligned}
\tag{12}
$$

where $\alpha_1$, $\alpha_{11}$, $\alpha_{12}$, $\alpha_{111}$, $\alpha_{112}$, and $\alpha_{123}$ are Landau energy coefficients, whose values determine the thermodynamic behaviors of the bulk. In our simulation work, the difference between the PMN–30PT matrix and the BT plate are reflected by the difference in $f_{bulk}$.

The gradient energy density is expressed as:

$$
\begin{aligned}
f_{grad} =\ & \frac{1}{2} G_{11}\left(P_{1,1}^2 + P_{2,2}^2 + P_{3,3}^2\right) + G_{12}\left(P_{1,1} P_{2,2} + P_{2,2} P_{3,3} + P_{1,1} P_{3,3}\right) \\
& + \frac{1}{2} G_{44}\left[\left(P_{1,2} + P_{2,1}\right)^2 + \left(P_{2,3} + P_{3,2}\right)^2 + \left(P_{1,3} + P_{3,1}\right)^2\right] \\
& + \frac{1}{2} G_{44}'\left[\left(P_{1,2} - P_{2,1}\right)^2 + \left(P_{2,3} - P_{3,2}\right)^2 + \left(P_{1,3} - P_{3,1}\right)^2\right]
\end{aligned}
\tag{13}
$$

where $G_{ij}$ are gradient energy coefficients. $P_{i,j}$ denote $\partial P_i/\partial r_j$.

The corresponding elastic energy densities are expressed as:

$$f_{elas} = \frac{1}{2} c_{ijkl} e_{ij} e_{kl} = \frac{1}{2} c_{ijkl} (\varepsilon_{ij} - \varepsilon_{ij}^0)(\varepsilon_{kl} - \varepsilon_{kl}^0) \quad (14)$$

where $c_{ijkl}$ is the elastic stiffness tensor, $\varepsilon_{ij}$ the total strain, $\varepsilon_{kl}^0$ the electrostrictive strain, i.e., $\varepsilon_{kl}^0 = Q_{ijkl} P_k P_l$.

The electrostatic energy density is given by:

$$f_{elec} = -\frac{1}{2} E_i^{in} P_i - E_i^{ex} P_i \quad (15)$$

where $E_i^{in}$ is the E-field induced by the dipole moments in the specimen. The detailed expression of $E_i^{in}$ is described in ref. [35]. $E_i^{ex}$ is an applied external E-field.

In computer simulation, a semi-implicit Fourier-spectral method is adopted for numerically solving the TDGL equation[36]. For BT template, the Landau free energy parameters were adopted from ref. [37]: $\alpha_1 = 3.34 \times 10^5 \times (T\text{-}381)$ C$^{-2}$ m$^2$ N, $\alpha_{11} = 4.69 \times 10^6 \times (T\text{-}393) - 2.02 \times 10^8$ C$^{-4}$ m$^6$ N, $\alpha_{12} = 3.23 \times 10^8$ C$^{-4}$ m$^6$ N, $\alpha_{111} = -5.52 \times 10^7 \times (T\text{-}393) + 2.76 \times 10^8$ C$^{-6}$ m$^{10}$ N, $\alpha_{112} = 4.47 \times 10^9$ C$^{-6}$ m$^{10}$ N, and $\alpha_{123} = 4.91 \times 10^9$ C$^{-6}$ m$^{10}$ N. For PMN-30PT matrix, the Landau free energy parameters were adopted from ref. [38]: $\alpha_1 = 0.745 \times (T\text{-}385.7) \times 10^5$ C$^{-2}$ m$^2$ N, $\alpha_{11} = -0.5 \times 10^8$ C$^{-4}$ m$^6$ N, $\alpha_{12} = -0.5125 \times 10^8$ C$^{-4}$ m$^6$ N, $\alpha_{111} = 0.5567 \times 10^9$ C$^{-6}$ m$^{10}$ N, $\alpha_{112} = 1.333 \times 10^9$ C$^{-6}$ m$^{10}$ N, and $\alpha_{123} = 0.24 \times 10^9$ C$^{-6}$ m$^{10}$ N. The electrostrictive coefficients and elastic constants of BT were set to be: $Q_{11} = 0.10$ m$^4$ C$^{-2}$, $Q_{12} = -0.034$ m$^4$ C$^{-2}$, $Q_{44} = 0.029$ m$^4$ C$^{-2}$, $s_{D\,11} = 9.07 \times 10^{-12}$ m$^2$ N$^{-1}$, $s_{D\,12} = -3.186 \times 10^{-12}$ m$^2$ N$^{-1}$, $s_{D\,44} = 8.197 \times 10^{-12}$ m$^2$ N$^{-1}$ (refs. [35,37]). The electrostrictive coefficients and elastic constants of PMN-0.30PT were set to be: $s_{D\,11} = 20 \times 10^{-12}$ m$^2$ N$^{-1}$, $s_{D\,12} = -7.5 \times 10^{-12}$ m$^2$ N$^{-1}$, $s_{D\,44} = 20 \times 10^{-12}$ m$^2$ N$^{-1}$, $Q_{11} = 0.055$ m$^4$ C$^{-2}$, $Q_{12} = -0.023$ m$^4$ C$^{-2}$, $Q_{44} = 0.03$ m$^4$ C$^{-2}$ (refs. [38–40]). Three-dimensional $64 \times 64 \times 32$ discrete grid points and periodic boundary conditions were employed. The mechanical boundary condition of the simulation is stress-free condition. The grid space in real space was $\Delta x = \Delta y = \Delta z = 1$ nm. The gradient energy coefficients are chosen to be $G_{11}/G_{110} = 1.5$, $G_{12}/G_{110} = 0$, $G_{44}/G_{110} = 0.75$, where $G_{110} = 7.04 \times 10^{-11}$ C$^{-2}$ m$^4$ N. Based on these parameters, the simulated width of domain walls was found to be 2 nm, which is consistent with experimental results of perovskite ferroelectrics[41]. It is worth noting that we also did phase-field simulations by using the Landau parameters of BT reported in ref. [42], and the similar results are obtained.

## Data availability

The data that support the findings of this study are included with the manuscript as Supplementary Information. Any other relevant data are also available upon request from F.L.

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

## Acknowledgements

F.L. and J.L. acknowledge the support of the National Natural Science Foundation of China (Grant Nos. 51922083, 51831010, and 51802182). Y.C. thanks the support of National Natural Science Foundation of China (11572103) and the Natural Science Foundation of Heilongjiang Province (YQ2019E026). The authors would like to thank Dr. Xiaoming Chen and Miss Xingxing Wang for the help on the measurements of temperature-dependent piezoelectric coefficients.

## Author contributions

The work was conceived and designed by S.Y. and F.L.; S.Y. fabricated the samples and performed the microstructure experiments; S.Y. and L.Q. performed the electro-mechanical properties measurement; Y.L. and F.L. performed the phase-field simulations; M.W. and J.L. assisted the fabrication of templates and textured ceramics; H.D. and X.G. assisted the properties measurements for the samples; F.L., J.L., Y.C., and Z.X. supervised the fabrication and test of the samples; F.L. and S.Y. drafted the manuscript; S.Z. revised the manuscript; and all authors discussed the results.

## Competing interests

The authors declare no competing interests.
