## [Peer Review File · Nature Communications]

REVIEWER COMMENTS

Reviewer #1 (Remarks to the Author):

The manuscript titled "Textured Ferroelectric Ceramics with High Electromechanical Coupling Factors over a Broad Temperature Range" is written very well and arranged in precise manner. The manuscript consists of worthy data justified with the results and discussion. However, I didn't find any merit to be considered in Nature Communications. The manuscript is more suited to Scientific Reports, Applied Physics Letter or the journal which is associated with the core solid state chemistry/physics such as Journal of Solid State Chemistry, American or European Ceramic Society Journals.

Reviewer #2 (Remarks to the Author):

The advancement of TGG PIN-PSN-PT ceramics is not clearly explained. If possible, please add a figure with the results of this work and other materials of single crystal for comparison. With high electromechanical coupling factor and high Curie temperature, authors can introduce a FOM (figure of merit) for the operation of piezoelectric and ferroelectric transducer.

Explain the relation between ferroelectric polarization direction and crystal orientation dependent piezoelectric properties. The polarization direction of PIN-PSN-PT ceramic is (111), then why not try to make <111> oriented ceramics. Do you expect <111> oriented ceramics have better piezoelectric properties than <100> oriented ceramics have.

Discuss the possibility of 'reactive' TGG process. How much BT will be reacted with 7 vol.% BT templated ceramics? This manuscript do not consider reaction of template with high sintering process.

In figures

In Fig. 1d, is F_{001} really 0 for R-0BT? The intensities of peaks are some different from the JCPDS powder data, so that F_{001} could be negative value. Please calculate with real experimental data. Even in random ceramics, XRD pattern is not always same as that of powder.

In Fig. 1f, the data were obtained with fracture surface or sample surface?

In Fig. 2e,f,g,h, Dielectric permittivity -> Dielectric constant and Loss factor -> Dielectric loss ($\tan\delta$)

Minor corrections

1. BT and BaTiO₃ are used simultaneously.
2. Dielectric permittivity -> dielectric constant
3. Why Pb₃O₄ is used for Pb source?
4. In the definition of Lotgering method of Eq. (3), please check denominator of P_0 ; $I_0(00l)$ -> $I_0(hkl)$.
5. Delta f is not defined in Eqs. (5) and (6).

Reviewer #3 (Remarks to the Author):

The work by S. Yang et al. reported ultrahigh electromechanical coupling factors over a broad temperature range in PIN-PSN-PT textured ceramics. Textured piezoelectric ceramics have been studied for more than twenty years, as the authors pointed out in the introduction, however, the reduced Trt as a result of the addition of the BaTiO₃ and SrTiO₃

templates severely restricted the practical applications of textured piezoelectric ceramics. In this work, the authors presented a solution to this long-standing technical challenge. To my knowledge this is the first relaxor-PT textured ceramics with refractory scandium oxide, and I understand the motivation of this research, which is expected to increase the temperature usage range. They successfully fabricated the PIN-PSN-PT textured ceramics by using a very simple but never reported sintering aids, i.e., the combination of B_2O_3 and CuO , and the Lotgering factor is as high as 99%. Specifically, the new PIN-PSN-PT textured ceramics showed temperature insensitive electromechanical coupling factor k_{33} of 88~89% from room temperature to 170°C. The electromechanical properties of PIN-PSN-PT textured ceramics are comparable to that of the state-of-the-art relaxor-PT crystals, while with much broader temperature usage range, which is supposed to greatly benefit many advanced transducers, for example, nondestructive evaluation testing and medical imaging transducers.

In my opinion, the electromechanical property of the textured PIN-PSN-PT ceramics is very impressive to the ferroelectric community. The experiments on piezoelectric properties and structural evolution as a function of temperature are well designed and solid. The authors also gave theoretical phase-field modeling results to explain the variation of phase transition temperature with addition of $BaTiO_3$ templates, which is a challenge for textured ceramics. The manuscript is well written and has an appropriate length. In principle, I'm in favor of the publication of this work in Nature Communications due to the significant advance in the performance of piezoelectric materials and the simple fabrication method with expected low manufacturing cost. While, before the acceptance of this manuscript, the following comments must be addressed by the authors.

1. In the method section, the authors discussed that the coupling factor k_{33} was determined by two methods (from the empirical engineering equation based on k_t and k_p , or based on the clamped and free dielectric permittivities). In my opinion, it is better to determine the k_{33} directly from a "33-bar" by resonance method based on the IEEE Standard on Piezoelectricity. I understand that it might be difficult to have the textured ceramic samples with enough thickness by tape casting method to make the 33-bar. While, I still encourage the authors to do this work, since this should be a very important progress in the ferroelectric research field, if the authors could measure the k_{33} directly from the 33-bar (as far as I know, there wasn't any reference reporting the k_{33} of textured ceramics from 33-bar).
2. Following the first question, can the authors give the details about the method to measure the planar mode coupling factor k_p ? I knew there were couple references reporting very high coupling k_p up to 90% which made no sense. It will be good for the readers to follow up the correct measurement method and calculation equation for textured ceramics which are certainly different from ceramics, to report the correct and real values without any misleading.
3. Did the authors measure the electric-field-induced strains for the textured ceramics with respect to temperature? Is the variation of strains with respect to temperature the same to that of piezoelectric coefficients? For phase-field simulations, the authors used PMN-PT as an example. The question is that why the authors did not perform phase field simulation directly on the PIN-PSN-PT system?

Responses to Reviewers' comments

and the description of revisions in the revised manuscript and supplementary materials

Dear Editor and Referees:

We would like to first express our best wishes for health and safety to you all in these uncertain times. We sincerely thank the reviewers for their time and effort in carefully reading the manuscript and in preparing the review reports. We have revised our manuscript accordingly, and we believe its quality is greatly improved. The point-by-point responses to comments are enclosed. We hope we have satisfactorily addressed all reviewers' concerns and questions.

Referees' comments and authors' replies:

[Referees' comments are in black; Author responses are in blue; Revisions in the manuscript are highlighted.]

Response to reviewer #1

Comment: The manuscript titled "Textured Ferroelectric Ceramics with High Electromechanical Coupling Factors over a Broad Temperature Range" is written very well and arranged in precise manner. The manuscript consists of worthy data justified with the results and discussion. However, I didn't find any merit to be considered in Nature Communications. The manuscript is more suited to Scientific Reports, Applied Physics Letter or the journal which is associated with the core solid state chemistry/physics such as Journal of Solid State Chemistry, American or European Ceramic Society Journals.

Reply: We thank the reviewer for considering our manuscript "very well written and contains worthy data". Based on the reviewer's comments, we would like to clarify the novelty and significance of this research, to justify the publication in *Nature Communications*.

(1) We reported the first textured relaxor-PT ferroelectric ceramic exhibiting clear advantages compared to state-of-the-art PMN-PT and PIN-PMN-PT crystals

The low T_{rt} as a result of the presence of BaTiO₃ or SrTiO₃ templates has been one of the most critical issues for the textured relaxor-PT ceramics, which greatly reduced the advantages over their crystal counterparts and conventional piezoelectric ceramics. In our present work, the newly designed <001>-textured PIN-PSN-PT ceramic is the first piezoelectric material that possesses k_{33} larger than 85%, k_p larger than 80% meanwhile with operational temperature (i.e., T_{rt}) up to 200°C, together with the enhanced coercive field (E_c) being around 7.5 kV/cm, showing great advantage compared to PMN-PT and PIN-PMN-PT crystals (T_{rt} : 60~120°C; E_c : 2~4 kV/cm). In addition to the enhanced T_{rt} , E_c and high coupling factors, the high compositional uniformity thus high property homogeneity, and low manufacturing cost compared to single crystals grown from melt, making the <001>-textured PIN-PSN-PT ceramics beneficial to numerous advanced piezoelectric transducers and sensors.

(2) We demonstrated a successful case to fabricate textured perovskite ceramics that contain a large amount of refractory components

Although the addition of Sc₂O₃ has the ability to increase the T_{rt} of the system, it causes great

difficulty to the template-induced grain growth process. To address this problem, we put considerable efforts in the fabrication process, and found that a small amount of CuO and B₂O₃ (0.2~0.5 wt%) could effectively assist the template-induced grain growth in PIN-PSN-PT system. Based on this discovery, we successfully synthesized high quality <001>-textured PIN-PSN-PT ceramics with the texturing degree (i.e., Lotgering factor F_{001}) of 99%. This method is expected to be used for other perovskite systems, for example PbZrO₃-PbTiO₃ (T_{ft} could be higher than 300°C, but refractory oxide ZrO₂ make it difficult to be made into textured ceramics), to explore new textured ceramics with even higher operational temperature range and high electromechanical properties.

As discussed above, we believe that the present work will rekindle the confidence and research enthusiasm of the ferroelectric community for investigating textured piezoelectric ceramics, and promote the development of next-generation piezoelectric ceramics.

Response to reviewer #2

Comment: The advancement of TGG PIN-PSN-PT ceramics is not clearly explained. If possible, please add a figure with the results of this work and other materials of single crystal for comparison. With high electromechanical coupling factor and high Curie temperature, authors can introduce a FOM (figure of merit) for the operation of piezoelectric and ferroelectric transducer.

Reply: We thank the reviewer for the valuable comment. According to this comment, we have compared the performance of state-of-the-art relaxor-PT single crystals and our textured ceramics (Fig.R1), which has been updated in Fig. 5d of the revised manuscript.

Fig. R1 The k_{33} as a function of T_c for a series textured PIN-PSN-PT ceramics and compared to relaxor-PT single crystals, where the k_{33} value range of state-of-the-art “soft” PZT ceramics is marked by the gray color. (Fig. 5d in the revised manuscript has been updated.)

We also thank the reviewer for suggesting us to introduce a FOM based on electromechanical coupling factor and Curie temperature. While, it is difficult to give a FOM based on these two factors, because the Curie temperature of piezoelectric material is a prerequisite for the operation of piezoelectric transducers. If the Curie temperature of a piezoelectric material is lower than the operation temperature of a transducer, the material cannot be used, no matter how high the electromechanical coupling is. The thumb of rule in practical piezoelectric transducer applications is to select the ferroelectric material with Curie temperature higher than the operational temperature, and then the material with high electromechanical coupling factor is desired. Thus, Fig. R1, the values of k_{33} against the T_c temperature, will be an important benchmark for comparing ferroelectric materials.

Comment: Explain the relation between ferroelectric polarization direction and crystal orientation dependent piezoelectric properties. The polarization direction of PIN-PSN-PT ceramic is (111), then why not try to make <111> oriented ceramics. Do you expect <111> oriented ceramics have better piezoelectric properties than <100> oriented ceramics have.

Reply: We appreciate the very valuable comment. To obtain the optimum piezoelectric coefficient d_{33}^* (* indicates that the direction ‘3’ is not along the spontaneous polarization), the rhombohedral relaxor-PT crystals are generally poled along [001] direction, though it shows the highest polarization along [111] direction. This phenomenon is due to the fact that in rhombohedral relaxor-PT crystals, the single domain shear coefficient d_{15} is much larger than the single domain longitudinal coefficient

d_{33} (i.e., $d_{15} \gg d_{33}$). Fig. R2 gives the orientation dependence of d_{33}^* for a rhombohedral relaxor-PT crystal, from which we can see that the maximum d_{33}^* is approximately along the [001] direction, being much larger than that along [111] direction [Liu X. *et al. Appl. Phys. Lett.*, 96, 012907 (2010)]. Therefore, we chose the PIN-PSN-PT system with rhombohedral composition and textured along the $\langle 001 \rangle$ direction, to achieve the optimum piezoelectric and electromechanical properties.

[Redacted]

Fig. R2 Orientation dependence of the piezoelectric coefficient d_{33}^* for a rhombohedral PIN-PMN-PT crystals. For plotting the figures, the X axis is fixed along $[1\bar{1}0]$ direction, the Z and Y axis are rotated around X axis. It can be seen that the maximum d_{33}^* is obtained when the Z' axis is rotated to the direction near the [001] direction. [Liu X. *et al., Appl. Phys. Lett.*, 96, 012907 (2010)]

Comment: Discuss the possibility of ‘reactive’ TGG process. How much BT will be reacted with 7 vol.% BT templated ceramics? This manuscript do not consider reaction of template with high sintering process.

Reply: Thanks for the very constructive comment. It is important to discuss the reaction of BT in textured ceramics.

According to this comment, we performed the SEM-EDS experiment for T-7BT sample to semi-quantitatively analyze the reaction between the templates and ceramic matrix. There are two types of templates: ‘thick’ templates and very thin templates. Fig. R3a shows an example for the ‘thick’ templates, whose thickness is about 1 μm , being within the thickness of the as-prepared templates. The line scanning element analysis of EDS shows that for these ‘thick’ templates there is a small amount of Ba diffusion in the ceramic matrix (Fig. R3b). On the other hand, there are also some very thin templates in the textured ceramics, whose thickness can be as low as 0.1 μm , being much smaller than the average thickness of the template (Fig. R3c). These templates were thought to be actively reacted with the ceramic matrix. In addition, we observed some holes and fissions around the thin template (Fig. R3c), being thought to be left after the reaction. The EDS results prove the significant diffusion of Ba from the thin template into the ceramic matrix (Fig. R3d).

Fig. R3 The SEM image and line scanning element analysis of EDS across BT and PIN-PSN-PT matrix in T-7BT sample. a-b, the SEM image and corresponding EDS line scanning element analysis of a ‘thick’ template; c-d, the SEM image and corresponding EDS line scanning element analysis of a severely eroded template. (Added as Supplementary Fig. 5 in the revised manuscript)

According to the reviewer’s suggestion, we added SEM-EDS results and discussed the reaction of templates in supplementary document of the revised manuscript:

“The EDS shows that for these ‘thick’ templates, a small amount of Ba diffusion was observed in the ceramic matrix (Supplementary Fig. 5b). On the other hand, there are also some very thin templates in the textured ceramics, as shown in Supplementary Fig. 5c. The thickness is much smaller than the average thickness of the templates, being thought to be actively reacted with the ceramic matrix. In addition, there are some holes and fissions were observed around the template, being thought to be left after the reaction. The EDS results prove the significant diffusion of Ba from the thin template into the ceramic matrix (Supplementary Fig. 5d).”

Comment: In Fig. 1d, is F_{001} really 0 for R-0BT? The intensities of peaks are some different from the JCPDS powder data, so that F_{001} could be negative value. Please calculate with real experimental data. Even in random ceramics, XRD pattern is not always same as that of powder.

Reply: Thanks for the valuable comment. We agree with the reviewer that the XRD pattern of random ceramics is not always the same as that of powders. Following the suggestion, we compared the XRD patterns of powders and ceramic (R-0BT), and we found that the F_{001} of R-0BT was -13.1% (Fig. R4a), indicating that the probability of $\langle 100 \rangle$ grains in the nontextured ceramic is lower than that in the powders. Based on this comment, we recalculated the texturing degree of textured ceramics (T-3BT, T-5BT and T-7BT), as show in Fig. R4b. We updated Fig. 1d in the revised manuscript.

Fig. R4 XRD patterns of the PIN-PSN-PT powders, random ceramic and the textured ceramics with 3 vol.%, 5 vol.% and 7 vol.% BT templates. Here, we calculate the F_{001} of the random ceramic and textured ceramics based on the XRD pattern of the powders.

Comment: In Fig. 1f, the data were obtained with fracture surface or sample surface?

Reply: Thanks for the question. In Fig. 1f, the EBSD data were obtained on sample surface. We added this description in the Fig. 1f of the revised manuscript.

Comment: In Fig. 2e,f,g,h, Dielectric permittivity -> Dielectric constant and Loss factor -> Dielectric loss ($\tan\delta$)

Reply: Thanks for the comment. Following the suggestion, we have replaced Dielectric permittivity with Dielectric constant, and Loss factor with Dielectric loss ($\tan\delta$).

Comment: Minor corrections

1. BT and BaTiO₃ are used simultaneously.

Reply: Thanks for the comment. We have corrected in the revised manuscript.

2. Dielectric permittivity -> dielectric constant

Reply: Thanks for the comment. In the revised manuscript, we changed “Dielectric permittivity” to “dielectric constant”.

3. Why Pb₃O₄ is used for Pb source?

Reply: Thanks for the comment. We didn’t have any special consideration on this issue. For the fabrication of Pb-based perovskite ceramics, the Pb₃O₄ and PbO are generally selected as Pb source. In this work, we used both Pb₃O₄ and PbO as Pb sources and didn’t find any difference in the textured PIN-PSN-PT ceramics.

Comment: 4. In the definition of Lotgering method of Eq. (3), please check denominator of P_0 : $I_{0(00l)} \rightarrow I_{0(hkl)}$. Delta f is not defined in Eqs. (5) and (6).

Reply: We thank the reviewer for the very careful reading. Delta f is equal to $(f_a - f_r)$. Corrections have been made in the revised manuscript, as shown in below:

$$F_{00l} = \frac{P - P_0}{1 - P_0}, P = \frac{\sum I_{(00l)}}{\sum I_{(hkl)}}, P_0 = \frac{\sum I_{0(00l)}}{\sum I_{0(hkl)}} \quad (3)$$

$$\frac{k_{31}^2}{1 - k_{31}^2} = \frac{\pi f_a}{2 f_r} \tan\left(\frac{\pi f_a - f_r}{2 f_r}\right) \quad (5)$$

$$k_t^2 = \frac{\pi f_r}{2 f_a} \tan\left(\frac{\pi f_a - f_r}{2 f_a}\right) \quad (6)$$

Response to reviewer #3

Comment: The work by S. Yang et al. reported ultrahigh electromechanical coupling factors over a broad temperature range in PIN-PSN-PT textured ceramics. Textured piezoelectric ceramics have been studied for more than twenty years, as the authors pointed out in the introduction, however, the reduced T_{rt} as a result of the addition of the $BaTiO_3$ and $SrTiO_3$ templates severely restricted the practical applications of textured piezoelectric ceramics. In this work, the authors presented a solution to this long-standing technical challenge. To my knowledge this is the first relaxor-PT textured ceramics with refractory scandium oxide, and I understand the motivation of this research, which is expected to increase the temperature usage range. They successfully fabricated the PIN-PSN-PT textured ceramics by using a very simple but never reported sintering aids, i.e., the combination of B_2O_3 and CuO , and the Lotgering factor is as high as 99%.

Specifically, the new PIN-PSN-PT textured ceramics showed temperature insensitive electromechanical coupling factor k_{33} of 88~89% from room temperature to 170°C. The electromechanical properties of PIN-PSN-PT textured ceramics are comparable to that of the state-of-the-art relaxor-PT crystals, while with much broader temperature usage range, which is supposed to greatly benefit many advanced transducers, for example, nondestructive evaluation testing and medical imaging transducers.

In my opinion, the electromechanical property of the textured PIN-PSN-PT ceramics is very impressive to the ferroelectric community. The experiments on piezoelectric properties and structural evolution as a function of temperature are well designed and solid. The authors also gave theoretical phase-field modeling results to explain the variation of phase transition temperature with addition of $BaTiO_3$ templates, which is a challenge for textured ceramics. The manuscript is well written and has an appropriate length. In principle, I'm in favor of the publication of this work in Nature Communications due to the significant advance in the performance of piezoelectric materials and the simple fabrication method with expected low manufacturing cost. While, before the acceptance of this manuscript, the following comments must be addressed by the authors.

Reply: We greatly appreciate the positive comments and recommendation by the reviewer.

Comment: In the method section, the authors discussed that the coupling factor k_{33} was determined by two methods (from the empirical engineering equation based on k_t and k_p , or based on the clamped and free dielectric permittivities). In my opinion, it is better to determine the k_{33} directly from a "33-bar" by resonance method based on the IEEE Standard on Piezoelectricity. I understand that it might be difficult to have the textured ceramic samples with enough thickness by tape casting method to make the 33-bar. While, I still encourage the authors to do this work, since this should be a very important progress in the ferroelectric research field, if the authors could measure the k_{33} directly from the 33-bar (as far as I know, there wasn't any reference reporting the k_{33} of textured ceramics from 33-bar).

Reply: Thanks for the valuable comments. Following this comment, we prepared T-3BT textured ceramic samples with sufficient thickness, and fabricated the 33-bars with size of $4 \times 1 \times 1 \text{ mm}^3$, as shown in Fig. R5a. Based on these 33-bars, we have determined the k_{33} for T-3BT textured ceramics. The k_{33} of T-3BT samples was calculated to be 87%, being consistent with the measurement results based on the other two methods.

Fig. R5 a, The picture of 33 mode samples of T-3BT; b, Impedance/phase spectra for longitudinal mode of T-3BT 33-bar. (updated Fig. 4b in the revised manuscript)

According to this comment, we have revised the corresponding sentences in revised manuscript:

“Three methods were used to determine the longitudinal coupling k_{33} : (1) based on the couplings factors k_p and k_t , as shown in Eq. 7; (2) based on the free and clamped dielectric constants, as shown in Eq. 8; (3) measured from the 33-bars with size of $4 \times 1 \times 1 \text{ mm}^3$, by using Eq. 9 based on IEEE Standard on Piezoelectricity.

$$k_{33}^2 \approx k_p^2 + k_t^2 - k_p^2 \cdot k_t^2 \quad (7)$$

$$k_{33} = \sqrt{1 - \frac{\varepsilon_{33}^S}{\varepsilon_{33}^T}} \quad (8)$$

$$k_{33}^2 = \frac{\pi f_r}{2 f_a} \tan\left(\frac{\pi f_a - f_r}{2 f_a}\right) \quad (9)$$

where ε_{33}^S is the clamped dielectric constant, ε_{33}^T the free dielectric constant, f_r the resonant frequency and f_a the anti-resonant frequency. The k_{33} calculated by the three methods are very similar with a variation below 2%.”

Comment: Following the first question, can the authors give the details about the method to measure the planar mode coupling factor k_p ? I knew there were couple references reporting very high coupling k_p up to 90% which made no sense. It will be good for the readers to follow up the correct measurement method and calculation equation for textured ceramics which are certainly different from ceramics, to report the correct and real values without any misleading.

Reply: We thank the reviewer for the comment. To calculate the k_p , we follow the method reported in the reference [Meitzler A. H. *et al.*, *IEEE Trans. Son. Ultrason.*, 20, 233-239 (1973)]. The calculation method of k_p is detailed in the following. First, we determined the Poisson’s ratio (σ) by the ratio of first overtone (f_{r1}) to fundamental resonant (f_r) frequencies. Second, the k^p (planar radial piezoelectric coupling coefficient) was determined by σ and $(f_a - f_r)/f_r$ based on the reference [Meitzler A. H. *et al.*, *IEEE Trans. Son. Ultrason.*, 20, 233-239 (1973)]. Finally, the reported planar coupling factor k_p was calculated by the following equation:

$$k_p^2 = \frac{(k^p)^2}{[(1+\sigma)/2+(k^p)^2]}$$

Here, we list the σ , k^p and k_p for different textured samples (0.19PIN-0.445PSN-0.365PT), as shown in the Table R1.

Table R1. The parameters of textured 0.19PIN-0.445PSN-0.365PT ceramics in the process of calculating k_p . The R-0BT, T-1BT, T-3BT, T-5BT and T-7BT indicate random, 1 vol.% BT 3 vol.% BT textured, 5 vol.% BT textured and 7 vol.% BT textured 0.19PIN-0.445PSN-0.365PT ceramics. It should be noted that the k^p is no upper bound exists. (Added as Supplementary Table. 2 in the revised manuscript).

Sample	f_r (kHz)	f_a (kHz)	f_{r1} (kHz)	σ	k^p (%)	k_p (%)
T-0BT	217.6	244.6	560.8	0.38	50.5	51.9
T-1BT	172.0	230.5	471.5	0.15	85.0	74.5
T-3BT	203.3	300	565.7	0.11	107.9	82.2
T-5BT	155.5	236	444.9	0.03	108.9	83.4
T-7BT	219.8	322.2	640.8	-0.02	95.2	80.5

According to the reviewer's suggestion, we added the following sentences and a formula to describe the calculation process of k_p in detail in the revised manuscript:

“Then, the k^p (planar radial piezoelectric coupling coefficient) was determined by σ and $(f_a-f_r)/f_r$ based on the Ref. 33. Finally, the coupling factor k_p was calculated by Eq. 4. All the parameters of textured 0.19PIN-0.445PSN-0.365PT ceramics for calculating k_p are listed in Supplementary Table 2.

$$k_p^2 = \frac{(k^p)^2}{[(1+\sigma)/2+(k^p)^2]} \quad (4)”$$

We have also noticed that some references reported inaccurate k_p of textured ceramics by using the following equation:

$$k_p = \frac{1}{\sqrt{0.395 \frac{f_r}{f_a-f_r} + 0.574}}$$

The problem is that this equation assumes that the Poisson's ratio (σ) of the textured ceramics is similar to that of conventional ceramics (for perovskite ferroelectrics, the value is in the range of 0.30~0.35). However, the fact is that σ of textured ceramic is much lower than that of conventional ceramics, as shown in Table R1.

Comment: Did the authors measure the electric-field-induced strains for the textured ceramics with respect to temperature? Is the variation of strains with respect to temperature the same to that of piezoelectric coefficients?

Reply: Thanks for the valuable comment. Following the suggestion, we measured the S-E curves of T-3BT sample at different temperatures from 20 to 230°C. The frequency and amplitude of the electric-fields are 1 Hz and 20 kV cm⁻¹, respectively. The strain increases monotonously below 150°C, and decreases significantly at 170°C as show in the Fig. R6, which is similar to the change of piezoelectric coefficient in the manuscript.

In the revised manuscript, we added the following statements:

“It is worth noting here that the electric-field-induced strain also increases monotonously from room

temperature to T_r , which is similar to the change of d_{33} , as shown in Supplementary Fig. 8.”

Fig. R6 Electric-field-induced strains of T-3BT at various temperatures. The frequency and amplitude of the electric-field are 1 Hz and 20 kV cm⁻¹, respectively. (Added as Supplementary Fig. 8 in the revised manuscript)

Comment: For phase-field simulations, the authors used PMN-PT as an example. The question is that why the authors did not perform phase field simulation directly on the PIN-PSN-PT system?

Reply: Thanks for the comment. The reason that we didn't perform phase-field simulation for PIN-PSN-PT is because that the phenomenological parameters of PIN-PSN-PT system are not available currently for simulation. In this work, we selected PMN-PT as a representative material to explore the general impact of BT on relaxor-PT ferroelectrics, since the phenomenological parameters of PMN-PT system have already been fitted in good agreement with the experimental data.

REVIEWERS' COMMENTS

Reviewer #1 (Remarks to the Author):

Authors have justified the query that I raised. I am convinced with the justification and recommend the manuscript for acceptance.

Reviewer #2 (Remarks to the Author):

The manuscript was much improved by revision. It can be published after minor corrections.

1. On dielectric constant, ϵ_{33} was used in content but ϵ_{33}/ϵ_0 was used in Tables. Actually dielectric constant = $\epsilon_r,33 = \epsilon_{33}/\epsilon_0$. I recommend $\epsilon_r,33$.
2. The temperature unit; K was used only in Fig. 3 but °C was used in other Figs. Please use °C in Fig. 3 for consistency.
3. Consider a reference; N. A. Pertsev et al. PRL 80(9) 1988 (1998) for Landau parameters for BT. The parameters are different from those in your reference and your calculation. Please explain the reason.

Reviewer #3 (Remarks to the Author):

I have read the revised manuscript carefully, I believe the authors have answered my comments and the answers sound reasonable. So I recommend acceptance.

Responses to reviewers' comments and the description of revisions in the revised manuscript

Dear Reviewers:

We sincerely thank you for the time and efforts on reviewing the manuscript. We have revised our manuscript accordingly. The point-by-point responses to comments are enclosed in the following.

Reviewers' comments and authors' replies:

[Reviewers' comments are in black; Author responses are in blue; Revisions in the manuscript are highlighted.]

Reviewer #1

Comment: Authors have justified the query that I raised. I am convinced with the justification and recommend the manuscript for acceptance.

Reply: We thank the reviewer for his/her positive recommendation and his/her efforts on reviewing the manuscript.

Reviewer #2

Comment: The manuscript was much improved by revision. It can be published after minor corrections.

Reply: We appreciate the reviewer for the positive recommendation. We have further revised our manuscript according to the reviewer's suggestions as described in the responses below.

Comment: 1. On dielectric constant, ϵ_{33} was used in content but ϵ_{33}/ϵ_0 was used in Tables. Actually dielectric constant = $\epsilon_{r,33} = \epsilon_{33}/\epsilon_0$. I recommend $\epsilon_{r,33}$.

Reply: We appreciate the reviewer for the careful reading and this suggestion. We have used ϵ_{33}^r to represent the dielectric constant in the revised manuscript.

Comment: 2. The temperature unit K was used only in Fig. 3 but °C was used in other Figs. Please use °C in Fig. 3 for consistency.

Reply: Thanks for the suggestion. In the revised manuscript, we have replaced K by °C in Fig. 3.

Comment: 3. Consider a reference; N. A. Pertsev et al. PRL 80(9) 1988 (1998) for Landau parameters for BT. The parameters are different from those in your reference and your calculation. Please explain the reason.

Reply: We thank the reviewer for the reference. As mentioned by the reviewer, there are several different groups of Landau parameters for BaTiO₃. However, the temperature-induced phase transition behaviors described by the different groups of Landau parameters are similar. Thus, to analyze the temperature-induced phase transition of the textured ceramics, we didn't have any special consideration on the selection of Landau parameters of BaTiO₃. According to this comment, we did phase-field simulation by using the parameters reported by N. A. Pertsev *et al.* As expected, we obtained the similar simulation results as that presented in the manuscript, i.e., the impact of 5

vol.% BaTiO₃ templates on the phase transition temperature of PMN-0.30PT is very small. In the revised manuscript, we added this reference and the following discussion.

“It is worth noting that we also did phase-field simulations by using the Landau parameters of BT reported in Ref. 42, and the similar results are obtained.”

Reviewer #3

Comment: I have read the revised manuscript carefully, I believe the authors have answered my comments and the answers sound reasonable. So I recommend acceptance.

Reply: We thank the reviewer for the positive recommendation and his/her efforts on reviewing the manuscript.